# Evaluation of Temperature Uniformity in a Middle-Refrigerated Truck Loaded with Pig Carcasses

**DOI:** 10.3390/foods12091837

**Published:** 2023-04-28

**Authors:** Hongwu Bai, Guanghong Zhou, Xianjin Liu

**Affiliations:** 1Key Laboratory of Meat Processing and Quality Control, Ministry of Education, College of Food Science and Technology, Nanjing Agricultural University, Nanjing 210095, China; hongwu.bai@jaas.ac.cn; 2Institute of Food Safety and Nutrition, Jiangsu Academy of Agricultural Sciences, Nanjing 210014, China; jaasliu@163.com

**Keywords:** refrigerated truck, computational fluid dynamics, carcass traceability, temperature field

## Abstract

In this study, we constructed a calculation model to determine the internal temperature field distribution in a medium-sized refrigeration truck with the dimensions of 4.1 m × 2.2 m × 2.2 m. Wind speed, air temperature, and carcass temperature were designated as the initial conditions. The k-ε model of computational fluid dynamics was used to simulate different wind speeds and ventilation duct settings on the carriage. Additionally, under specific boundary conditions, the speed of the air outlet, the types of ventilation ducts, and the carcass loads were all varied to determine the uniformity of the temperature field. The results showed that, when the air outlet speed was 5 m/s, the temperature field in the refrigerated truck was relatively more uniform. The simulated results were in good agreement with the measured results. The average absolute error was 0.35 °C, and the average relative error was 9.23%.

## 1. Introduction

Temperature is a key factor affecting the rapid propagation of microorganisms. If the temperature of a food storage area is not correct, then food spoilage will increase [1,2,3]. Refrigerated trucks are an important means of land transportation. During transportation, the internal temperature field within a refrigerated truck is affected by the external environment, internal structures, and the stacking methods used [4,5]. Many scholars have used computational fluid dynamics (CFD) to study refrigerated trucks, and the results of their work have helped to overcome the shortcomings of traditional theoretical analysis and enable simpler, more effective computation [6,7]. Scholars have also conducted numerical simulations of transport-compartment temperature fields based on airflow velocity, stacking patterns, and airflow organization [8]. Several temperature and humidity sensors in a refrigerated truck are usually managed at certain periods to simulate the corresponding temperature and humidity curves [4,9,10]. The analysis of turbulent flow with complex heat transfer in a refrigerated body was conducted by Kayansayan et al. [11]. Their study investigated the impact of the shape factors of the body by examining two types of cold air outlets, three types of aspect ratios, and the Reynolds number of the wind speed. The study concluded that the optimum cold-air speed value could be determined depending on the shape of the refrigerated body. However, to develop accurate modeling, experimental work should accompany these methods. Margeirsson et al. [12] developed a 3D heat transfer model that predicted changes in product temperature for supercooled cod fillets packaged in loaded EPS boxes. The model was validated by comparing with experimental results but was found to be slightly different from the actual results. In another study, Hoang [13] presented a refrigerated vehicle heat transfer model based on computational fluid dynamics (CFD) to predict the temperature of cargo. The CFD modeling considered various phenomena, such as the respiration heat of the product and infiltration of outside air due to the opening of the door. The results obtained reflect the influence of the thermal characteristics of the refrigerator compartment on the airflow structure, but they cannot truly reflect the convection heat transfer in the transport environment [14,15]. In the present study, 36 temperature and humidity nodes were installed in a passenger compartment to establish a wireless environmental detection system for refrigerated trucks and record the truck temperature in real time. The internal temperature field of the refrigerated truck was numerically simulated using Fluent 17 with the k-ε turbulence model. The distribution of the temperature field inside the truck was calculated under different wind speeds and ventilation modes. Finally, a temperature-based assessment model was ultimately developed, which could monitor the quality variations of pig carcasses during their transportation via refrigerated trucks [16].

## 2. Materials and Methods

### 2.1. Test Materials

In this work, we studied the temperature field of pig carcasses in a middle-refrigerated truck, because this type of truck is suitable for medium- and short-distance cold-chain transportation in big cities due to its low cost and widespread use. The middle-refrigerated truck had external dimensions of 4.3 m × 2.4 m × 2.4 m (length × width × height), an internal clearance of 4.1 m × 2.2 m × 2.2 m, a body thickness of 105 mm, and a covering made of Fiberglass reinforced plastic (FRP) with a thickness of 2.5 mm. The “sandwich” structure was a 100 mm polyurethane foam board with double doors at the back of the cabin, and the air outlet was placed directly in front of the carriage.

A refrigerated truck is not always fully loaded. Hence, it was necessary to compare the standard temperature fields with the same volume of pig carcass areas, to produce a credible evaluation model. The volumes of the standard temperature fields were equal. The carcass-volume ratio refers to the proportion between the volume of the carcass area being evaluated within the refrigerated compartment and the volume of the maximum allowable cargo area. An upper limit to the actual cargo volume of the refrigerated truck was specified and the carriage was not filled entirely, to ensure that airflow circulated in the interior part of the refrigerated truck. Figure 1 presents a schematic illustration of a large pig carcass area in a carriage with a standard temperature field. The selected area of the standard temperature field was used as the reference area, and the entire pig carcass area was reduced to a voided cube (Figure 1b). A constant distance between the pig carcass area and the inner wall of the carriage was assumed, and the height of the pig carcass area was unchanged. The distance from the top plate was termed *h*_1_ and the distance from the bottom plate was termed *h*_2_.
(1)$$x_{1}=x_{2}=y_{1}=y_{2}{}^{}$$
(2)$$V_{x}=\left(x_{0}-x_{1}-\Delta x_{2}\right)\times \left(y_{0}-y_{1}-y_{2}\right)\times {\left(h_{0}-h_{1}-h_{2}\right)}^{}\;\;$$

If *x*_1_, *x*_2_, *y*_1_, and *y*_2_ are 0, then
(3)$$V_{x}=V_{0}=x_{0}y_{0}h_{0}$$
(4)$$\eta =\frac{V_{x}}{V_{0}}$$

These formulas express the maximum volume of the standard temperature field, where η is the standard temperature field volume ratio. The length, width, and height (*h*_0_) of the compartment were all taken into consideration.

### 2.2. Test Conditions

The main unit (Testo 480) of a multi-functional environmental tester (Detu Instrument International Trading Co., Ltd., Shanghai, China) was used for the study. The probe parameters were as follows:Flexible wind-speed probe with a range of 0–20 m/s and a measurement accuracy of ±0.03 m/s;Surface-temperature probe with a range from −60 °C to 300 °C and a measuring accuracy of ±0.5 °C;Immersion-type PT100 temperature probe with a measuring range of −100 °C to 400 °C and a measuring accuracy of ±0.15 °C; and an air-humidity probe with a range of 0–100% RH and a measurement accuracy of ±1.8% RH.

An Internet-of-Things temperature and humidity node (Nanjing Huiming Instrument Co., Ltd., Nanjing, China) was used with the following parameters:Humidity measurement range of 0–100% RH;Humidity accuracy of ±1.8% RH with humidity resolution of ±0.05% RH;Temperature measurement range from −40 °C to 123.8 °C;Temperature accuracy of ±0.3 °C and temperature resolution of ±0.01 °C.

### 2.3. Test Methods

Many factors affect the temperature fields of refrigerated compartments. The main considerations in this study were the external temperature, the refrigerated vehicle ventilation mode, and the temperature of the air outlet in the compartment. These three factors, taken together, completely determine the standard temperature fields. Before each test, the temperature and humidity nodes were calibrated with a high-precision temperature and humidity sensor to ensure testing accuracy and consequently reduce testing error. In addition, the refrigerator truck was turned on for pre-cooling. During the test, the humidity node acquisition interval was set to 30 s, and the starting temperature of the vehicle compartment was determined. Wind speed has a transient instability value, so a wind-speed probe was affixed to the cold air outlet. The average wind speed under the maximum cooling capacity was used as the initial value. Before and after the testing, nine temperature and humidity nodes were arranged, at specific positions of 0.5, 2.0, and 4.0 m from the outlet location. Each test chain comprised four nodes suspended at a distance from the top of the carriage (i.e., 0.5, 1.0, 1.5, and 2.0 m). A temperature and humidity node was arranged on the outer wall of the carriage to measure outdoor temperature. The gateway, router, and GPRS transmission module were all located in the cab.

## 3. Models and Assumptions

### 3.1. Model Assumptions

To facilitate subsequent calculations, the following points were assumed for the model:The air in the compartment is a transparent radiation medium according to Boussinesq’s hypothesis.

The gas in the truck is a Newtonian fluid, and the pig carcass is considered to be a porous medium.

The effect of temperature changes on the physical parameters of air and swine fever can be ignored.

Any influence of pipes, rails, and wires in the refrigerated truck on the airflow can be ignored.

The walls of the truck body are insulated and the truck body is fully sealed, so that there is no leakage phenomenon.

The effects of moisture loss and latent heat of evaporation on the body temperatures of pig carcasses can be ignored.

The issue of heat dissipation in pig carcasses is not considered.

### 3.2. Basic Equations

In fluid heat transfer analysis, researchers often regard the mass conservation equation, momentum conservation equation, and energy conservation equation as the basic control equations [17]. The three basic equations of CFD can be referred to simply as follows:

mass conservation equation: (5)$$\frac{\partial \rho }{\partial t}+\frac{\partial }{\partial x_{i}}\left(\rho u_{i}\right)=S_{m}$$
conservation of momentum:(6)$$\frac{\partial }{\partial t}\left(\rho u_{i}\right)+\frac{\partial }{\partial x_{j}}\left(\rho u_{i}u_{j}\right)=\frac{\partial p}{\partial x_{i}}+\frac{\partial \tau_{ij}}{\partial x_{j}}+\rho g_{i}+F_{i}$$
energy conservation theorem:(7)$$\frac{\partial }{\partial t}\left(\rho E\right)+\nabla \cdot \left(\vec{v}\left(\rho E+\rho \right)\right)=\nabla \cdot \left(k_{eff}\nabla T-\underset{j}{\sum }h_{j}{\vec{J}}_{j}+{\bar{\tau }}_{eff}\cdot \vec{v}\right)+S_{h}$$
subsequently,
(8)$$\tau_{ij}=\left[\mu \left(\frac{\partial u_{i}}{\partial x_{j}}+\frac{\partial u_{j}}{\partial x_{i}}\right)\right]-\frac{2}{3}\mu \frac{\partial u_{l}}{\partial x_{l}}\delta_{ij}$$
(9)$$E=h+\frac{p}{\rho }+\frac{v^{2}}{2}$$
(10)$$h=\underset{j}{\sum }Y_{j}h_{j}$$
(11)$$h_{j}=\int_{T_{ref}}^{T}c_{p,j}dt$$
(12)$${\bar{\tau }}_{eff}=\mu_{eff}\left(\frac{\partial u_{j}}{\partial x_{i}}+\frac{\partial u_{i}}{\partial x_{j}}\right)-\frac{2}{3}\mu_{eff}\frac{\partial u_{i}}{\partial x_{i}}\delta_{ij}$$
where

*S_m_*—source item;*k_eff_*—effective thermal conductivity;*E*—total energy;*τ_ij_*—stress tensor;*Y_j_*—mass fraction of component;*T_ref_*—298.15 K;*τ_eff_*—partial stress equations;*x*—flow field direction;*t*—time;*ρ*—fluid density;*μ*—velocity vector;*p*—static pressure;*h*—height;*S*—fluid quality;*J*—fluid heat;*T*—force;*F*—surface force.

### 3.3. Thermal Conduction and Heat Convection

When the refrigerated truck is running on the road, the heat transfer of its compartment enclosure structure continues to change [18]. At the same time, the goods in the refrigerated truck are coupled to the heat transfer. Convection, conduction, and radiation all affect the temperature field, and the entire refrigerated truck is in a state of dynamic thermal equilibrium. The heat conduction equation of the inner and outer walls of the compartment can be expressed as follows:(13)$$q_{c}=\frac{t_{o}-t_{i}}{\frac{\delta_{i}}{\lambda_{i}}+\frac{\delta_{m}}{\lambda_{m}}+\frac{\delta_{o}}{\lambda_{o}}}$$
where

t_0_—temperature of the outer wall of the refrigerated compartment;t_i_—temperature of the wall surface of the refrigerated compartment;δ_i_—thickness of the FRP outside the refrigerator compartment;δ_m_—thickness of the polyurethane insulation board;δ_0_—thickness of the FRP in the refrigerated compartments;λ_i_ —λ_0_—thermal conductivity of the FRP of the refrigerated truck; λ_m_—thermal conductivity of polyurethane.

To investigate the refrigerated compartments during movement, we dynamically calculated the heat flow on the surfaces of the refrigerated trucks. The external heat flow mainly comes from solar radiant heat and the convection heat transfer between the inner and outer walls of the cabin. Because fresh pig carcasses are the object of this study, there is no respiratory heat in the animal tissue. The average heat flux is defined as the amount of heat that flows into the interior of a compartment per unit of time, per unit area of the wall that separates the compartment from the outside environment. Internal and external convection heat transfer must also be considered in addition to vehicle body heat transfer. The average heat flux density q_h_ of the refrigerated truck compartment wall can thus be expressed as:(14)$$q_{h}=\frac{t_{o}-t_{i}}{\frac{1}{h_{o}}+\frac{\delta_{m}}{\lambda_{m}}+\frac{1}{h_{i}}}$$
where

h_o_—convection heat transfer coefficient of the outer wall of the refrigerated compartment andh_i_—convection heat transfer coefficient of the inner wall of the refrigerated compartment.

The other parameters are the same as those in Equation (13). When the outer wall of the refrigerated truck is considered, the total heat transfer is, therefore:(15)$$Q_{c}=q_{c}\times A_{0}$$
where

A_0_—total area of the outer wall of the refrigerated compartment.

During transportation, the thermal load of the external thermal environment (i.e., heat transfer through the wall) accounts for more than 80% of the total heat load in the refrigerated truck. If the external temperature of the compartment is higher than the internal temperature, the heat-insulation layer of the compartment can only achieve a small amount of heat insulation. Within the unit area, when the difference between internal and external temperature increases, the heat from the external wall to the interior of the refrigerated compartment initially increases. Then, the evaporator inside the compartment fan spreads this heat to all parts of the compartment to form an unstable temperature field. Finally, the heat that enters the compartment exits through the truck’s cooling system. By using such means, a low-temperature environment can be maintained inside the compartment.

### 3.4. Temperature Field Uniformity Index

Refrigerated trucks in operation are not always fully loaded, and different cargoes may require different stacking methods. Therefore, the temperature field where a cargo is located should not be compared with the standard temperature field of the entire truck, but with the same volume as the stacking area [4]. The standard temperature field of the refrigerated truck is called the measured area, which is equal to the volume of this same area. In this study, the ratio of the cargo area volume to the maximum allowable cargo area volume is called the refrigerated truck volume. To ensure that airflow circulates in the interior, the actual capacity of the refrigerated truck is given an upper limit (i.e., it cannot fill the entire carriage). In other words, the maximum allowable cargo volume of the refrigerated truck is the same as the maximum reference volume of the standard temperature field.
(16)$$\mu_{0}=\sum_{i=1}^{n=4}\propto_{\begin{array}{c} i \end{array}}\eta^{i-1}\;\left(\mu_{\begin{array}{c} 0 \end{array}}=\propto_{0}+\propto_{1}\eta +\propto_{2}\eta^{2}+\propto_{3}\eta^{3}\right)$$
(17)$$\sigma_{0}=\overset{n=4}{\underset{i=1}{\sum }}\beta_{\begin{array}{c} i \end{array}}\eta^{i-1}\;(\sigma_{\begin{array}{c} 0 \end{array}}=\beta_{0}+\beta_{1}\eta +\beta_{2}\eta^{2}+\beta_{3}\eta^{3})$$
(18)$$R_{0}=\overset{n=4}{\underset{i=1}{\sum }}{\gamma \eta }^{i-1}\;(R_{0}=\gamma_{0}+\gamma_{1}\eta +\gamma_{2}\eta^{2}+\gamma_{3}\eta^{3})$$
(19)$$\eta =\frac{V_{x}}{V_{0}}$$
(20)μ1=∑i=1ntin
(21)σ1=∑i=1n(ti−μ)2/nμ
(22)$$R_{1}=\mathrm{Max}\left(t\right)-\min\left(t\right)$$
(23)$$\psi_{1}=\left(\mu -t_{0}\right)/(\mu_{0}-t_{0})$$
(24)ψ2=σσ0
(25)ψ3=RR0
(26)$$\xi =\psi_{1}+\psi_{2}+\psi_{3}$$
where

μ_0_—average temperature of the standard temperature field;ρ_0_—temperature non-uniform coefficient of the standard temperature field;R_0_—temperature of the standard temperature field;μ_1_—average temperature of the temperature field to be measured;ρ_1_—temperature non-uniform coefficient of the temperature field to be measured;R_1—_temperature of the temperature field to be measured;η—volumetric rate of the actual cargo carried with the standard temperature field in the refrigerated truck;α_i_—coefficient of the average temperature equation of the standard temperature field;β_i_—coefficient of the temperature non-uniform coefficient equation of the standard temperature field;γ_i—_coefficient of the temperature extreme equation of the standard temperature field;t_0_—outlet temperature;ψ_1—_average deviation from the temperature difference ratio;ψ_2_—temperature non-uniform coefficient ratio;ψ_3_—temperature range ratio;ξ—temperature field uniformity index.

The basic indicators (Formulas 16–18) used to evaluate the temperature field in the refrigerated truck are the average temperature of the temperature field, the temperature non-uniform coefficient, and the temperature range. The average deviation (Formulas 23–25) can be used to compare the temperature field with the standard temperature field and reveal any advantages or disadvantages. Formula 26 expresses the comprehensive inhomogeneity coefficient in the refrigerator truck.

### 3.5. Boundary Conditions

The bottom of the box is set as the adiabatic boundary condition;The air outlet adopts the speed boundary conditions;The returning-air inlets on both sides of the box are simplified using free-flow boundary conditions;The relevant physical parameters and initial boundary conditions used in the calculation model are shown in Table 1.

### 3.6. Simulation Conditions

The simulation was performed on a DELL R720 PC. The hardware parameters are as follows:

Xeon 6 core E5-2630 × 2 2.6 GHz; 6 × 16 G 1333 MHz; 3 × 300 G 3.5 15 K; RAID5; PERC H310; single 750 W; and a silent fan.

The k-ε model is used for calculations. The discrete solver based on pressure has a first-order upwind scheme for momentum, energy, turbulent kinetic energy, diffusion rate, and a SIMPLE algorithm for pressure–velocity coupling. The equations for turbulent kinetic energy transport are derived entirely, while the equations for dissipation rate are obtained through a combination of physical reasoning and mathematical simulations based on similar primitive equations. The model assumes that the flow is completely turbulent and that molecular viscosity can be ignored [19].

The turbulent kinetic energy k and dissipation rate ε in the standard k-ε model [20] are calculated as follows:(27)$$\rho \frac{Dk}{Dt}=\frac{\partial }{\partial x_{i}}\left[\left(\mu +\frac{\mu_{t}}{\sigma_{k}}\right)\frac{\partial k}{\partial x_{i}}\right]+G_{k}+G_{b}-\rho \varepsilon -Y_{M}$$
(28)$$\rho \frac{D\varepsilon }{Dt}=\frac{\partial }{\partial x_{i}}\left[\left(\mu +\frac{\mu_{t}}{\sigma_{k}}\right)\frac{\partial \varepsilon }{\partial x_{i}}\right]+C_{1\varepsilon }\frac{\varepsilon }{k}\left(G_{k}+C_{3\varepsilon }G_{b}\right)-C_{2\varepsilon }\rho \frac{\varepsilon^{2}}{k}$$
where

*G_k_*—turbulent kinetic energy due to the average velocity gradient;*G_b_*—a parameter used to generate turbulent kinetic energy due to buoyancy effects;*Y_M_*—effect of turbulent pulsation expansion on total dissipation rate;*μ_t_*—turbulent viscosity coefficient;*D_t_*—total derivative of time;*D_ε_*—the turbulent Prandtl numbers of ε;*D_k_*—the turbulent Prandtl numbers of k;*C*—constant.


(29)
$$\mu_{t}=\rho C_{\mu }\frac{k^{2}}{\varepsilon }$$


In Fluent software, *C*_1*ε*_ = 1.44, *C*_1*ε*_ = 1.92, *C_μ_* = 0.09, *σ_k_* = 1.0, and *σ_ε_* = 1.3. These default values are determined experimentally in basic turbulent flows, including commonly encountered shear flows, such as boundary layers, mixed layers, and jets, and attenuated isotropic mesh turbulence [21,22].

## 4. Results and Analysis

### 4.1. Flow Field Status Judgment

The laminar and turbulent flow states are based on the Reynolds number, which is determined in Equation (30). When the fluid sweeps the plate, the critical Reynolds number is 5 × 10^5^. If the actual Reynolds number is less than the critical Reynolds number [23], then the flow state is laminar. If the actual Reynolds number is greater than the critical Reynolds number, then the flow state is turbulent.
(30)Re=σVLμ
where

Re—Reynolds number;ρ—air density;V—airflow velocity at the outlet (m/s);L—flow distance of the airflow (m); μ—kinematic viscosity of the air (m^2^/s).

The air density at 0 °C is 1.293 kg/m^3^, the air kinematic viscosity is 13.28 × 10^−6^ m^2^/s [24], and the flow distance of the airstream is 1.5 m. Using Formula (31), the critical velocity in the empty truck is, therefore, calculated to be 0.125 m/s, as follows:(31)V=Re×μσ×L=50,000×13.28×10−61.293×4.1=0.125

The flow field along the four walls has a direct impact on the temperature field inside the truck. The turbulent airflow near the front, rear, left, or right walls can lead to an uneven temperature field near the inner wall of the cabin. On the other hand, laminar airflow near the walls can create a stable laminar boundary layer near the inner wall surface, resulting in an improved steady-state effect of the mainstream flow field. Figure 2 illustrates this phenomenon.

### 4.2. Influence of Wind Speed on Temperature Field

The simulated operating conditions are as follows: external temperature 35.0 °C; truck speed 20.8 m/s (at approximately 75.0 km/h); outlet velocity 1.0 m/s; and temperature 0 °C. The result becomes stable and steady after 12 min of iterative calculation. From the central axis (z = 1.1 m) to the end of the refrigerated truck, the air outlet velocity gradually declines from 1 m/s to 0.1 m/s, and the temperature rises from 2 °C to 10 °C. As shown in Figure 3, the average temperature is high, and most of the areas are red. When the wind speed is gradually increased to 3 m/s, the average temperature tends to fall, and only half of the area is red. When wind speed rises to 5 m/s, the overall temperature becomes uniform (i.e., the blue region dominates), and the average temperature is close to the outlet temperature. An increase in wind speed to 7 m/s can further improve the homogeneity of the temperature inside the compartment. However, it is worth noting that high-wind velocity can cause carcasses to become darker and drier, ultimately resulting in a lower commodity value. Due to the potential negative impact of high-wind speeds on carcass quality, a wind speed of 5 m/s may be considered the optimal choice, even though higher wind speeds can result in a more uniform temperature distribution within the refrigerated truck.

### 4.3. Effect of Ventilation Ducts on Temperature Field

Three air-supply ducts are installed on the top of the refrigerated truck joining with the air outlet. The main air-supply duct leads directly to the rear of the truck. The secondary air-supply duct leads to the middle part of the carriage. This design optimizes the airflow direction of the air supply, in order to improve the temperature uniformity of the refrigerated truck. When v = 1 m/s, the main area of the original truck appears mostly in red, which means that the high-temperature zone dominates. When the upper ventilation duct is added, the cold air is directly delivered to the middle and tail, and the efficiency of the air-conditioning system appears to improve. The simulated result appears in light red. For the second working condition, 16 U-shaped air ducts are installed on the floor of the refrigerated truck directly; these are called lower ventilation ducts. Under this condition, the turbulent flow at the floor is enhanced and the bottom temperature zone becomes greener. The third working condition involves the simultaneous operation of the upper and lower ventilation ducts. Under this condition, the temperature field becomes uniform over the same simulation time, and the average temperature approximates the target temperature more ideally, which is shown in Figure 4.

### 4.4. Evaluation of Temperature Field Uniformity

In this study, we numerically simulated four kinds of operating conditions: an original refrigerated truck; a truck with an upper duct; a truck with a lower duct; and a truck with upper and lower ducts. An evaluation model of a standard temperature field in a middle-refrigerated truck loaded with carcasses was established using experimental and simulated data. According to Equation (4), the volumetric rate of the experimental carriage could be calculated as follows:(32)$$\eta =\frac{V_{x}}{V_{0}}=\frac{3.5\times 2.0\times 2.0}{4.1\times 2.2\times 2.2}=0.706$$

η—Volume ratio of refrigerated vehicle load;*V_x_*—Volume of pig carcasses loaded in a refrigerated truck;*V*_0_—headroom volume of refrigerated truck.

The relevant data were reviewed and the following variables were obtained: the coefficient α is the average temperature equation of the standard temperature field; the coefficient β is the coefficient of non-uniformity coefficient; and the coefficient γ is the temperature range difference equation. Based on the results obtained from Equations (16) to (18), the inconsistent coefficients of the measured temperature field (Equation (20)), average temperature (Equation (21)), and temperature difference (Equation (22)) were calculated. The other required values were derived by using the average temperature ratio, the non-uniform coefficient ratio, and the temperature range ratio of Equations (23)–(25). As shown in Table 2, a smaller numerical value indicates a closer approximation to the ideal standard temperature field for the truck.

### 4.5. Test Verification

Figure 5 shows a comparison of the simulation results with real-world data, as measured using 36 temperature sensors. It can be seen that the simulated temperature values are consistent with the experimental values, with an average absolute error of 0.35 °C and an average relative error of 9.23%. For the truck in the original state, the average absolute error is 0.27 °C and the average relative error is 11.98%. For the truck with an upper ventilation duct, the average absolute error is 0.20 °C and the average relative error is 6.18%. For the truck with a lower ventilation duct, the average absolute error is 0.56 °C and the average relative error is 11.98%. For the truck with both upper and lower ventilation ducts, the average absolute error is 0.45 °C and the average relative error is 8.25%.

## 5. Conclusions

For this study, a typical middle-refrigerated truck was used, with a clearance of 4.1 m × 2.2 m × 2.2 m. Firstly, using experimental and simulated data, we established an evaluation model for a standard temperature field in a middle-refrigerated truck loaded with pig carcasses. We found that the greater the air-supply speed, the shorter the average temperature stabilization time. Secondly, we established a temperature field evaluation system for refrigerated trucks loaded with pig carcasses. To evaluate the performance of the refrigerated truck’s temperature field, parameters such as the average temperature, temperature range, and non-uniform coefficient were considered. By assigning appropriate weight coefficients to these parameters, the average deviation temperature ratio, temperature non-uniform coefficient ratio, and temperature range ratio were combined to produce a uniformity index for the temperature field inside the refrigerated truck. Finally, we studied how the use of upper and/or lower ventilation ducts affected the temperature field in the compartment of the refrigerated truck. The study found that incorporating three upper ventilation ducts and sixteen lower ventilation ducts led to an improvement in the uniformity of the temperature field inside the refrigerated truck. The results showed that the condition with 16 lower ventilation ducts was more effective, possibly because it allowed for a sufficient airflow return path in a middle-refrigerated truck that was almost fully loaded with pig carcasses.

## Figures and Tables

**Figure 1 foods-12-01837-f001:**
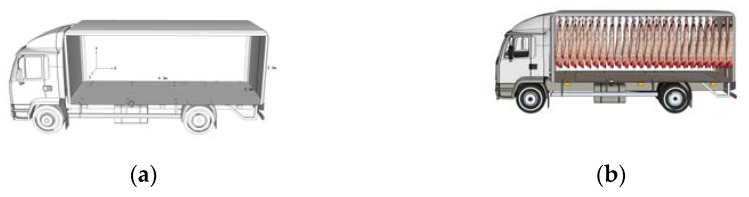
Structure of a middle-refrigerated truck and area occupied by pig carcasses. (**a**) An empty compartment. (**b**) A compartment with pig carcasses.

**Figure 2 foods-12-01837-f002:**
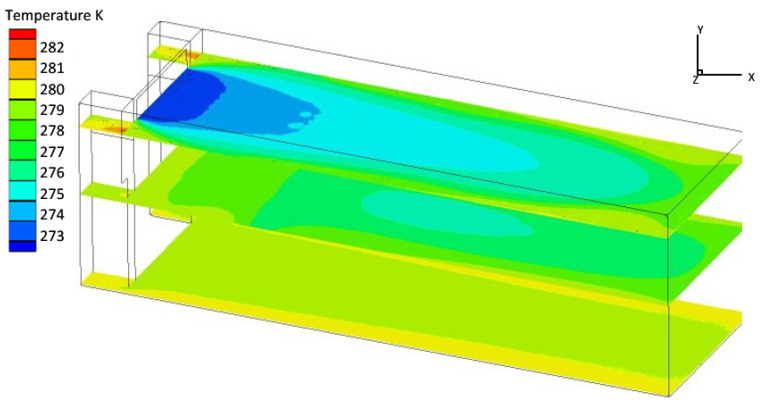
The standard temperature contour in a refrigerated truck with low-speed air supply.

**Figure 3 foods-12-01837-f003:**
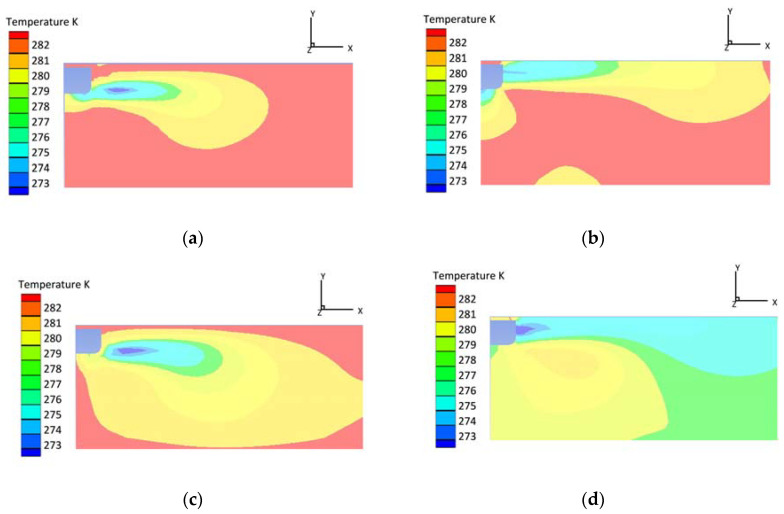
Temperature contours with different wind speeds (t = 0.1 h, z = 1.1 m): (**a**) wind speed (v = 1 m/s); (**b**) wind speed (v = 3 m/s); (**c**) wind speed (v = 5 m/s); and (**d**) wind speed (v = 7 m/s).

**Figure 4 foods-12-01837-f004:**
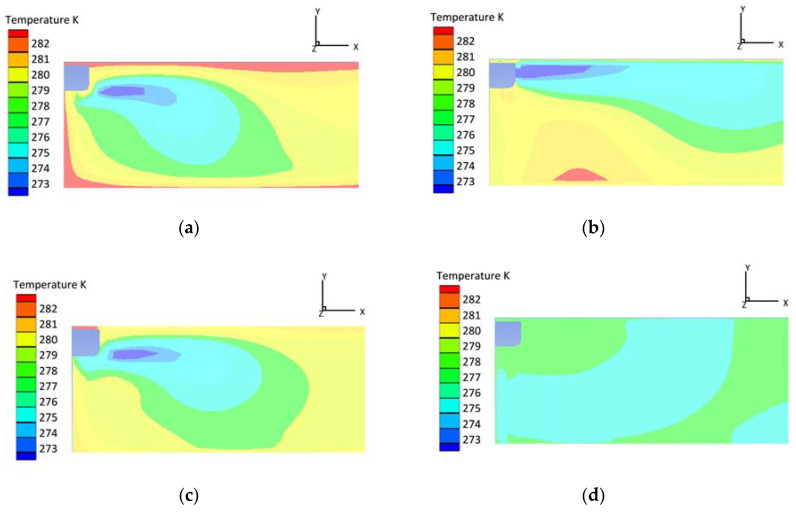
Temperature contours with different ventilation ducts: (**a**) original refrigerated truck (v = 1 m/s, h = 0.1); (**b**) with added upper duct (v = 1 m/s, h = 0.1); (**c**) with added lower duct (v = 1 m/s, h = 0.1); and (**d**) with added upper and lower ducts (v = 1 m/s, h = 0.1).

**Figure 5 foods-12-01837-f005:**
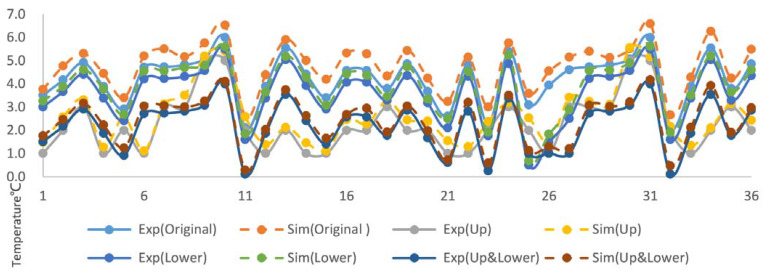
Experimental and simulated temperature values for different operating conditions. Exp(Original): data measured on the original unmodified refrigerated truck; Exp(Lower): data measured on a refrigerated truck added lower ventilation ducts; Sim(Original): the simulated values on the original unmodified refrigerated truck; Sim(Lower): the simulated values on a refrigerated truck added lower ventilation ducts; Exp(Up): data measured on a refrigerated truck added up ventilation ducts; Exp(Up and Lower): data measured on a refrigerated truck added both up and lower ventilation ducts; Sim(Up): the simulated values on the refrigerated truck added up ventilation ducts; Sim(Up and Lower): the simulated values on the refrigerated truck added both up and lower ventilation ducts.

**Table 1 foods-12-01837-t001:** Physical parameters and initial boundary conditions.

Parameter	Initial Value	Unit	Instructions
Density of FRP	2100	kg/m^3^	Merchant
Thickness of FRP	0.0025	m	Merchant
Density of polyurethane foam plate	40	kg/m^3^	FRT properties
Thickness of polyurethane foam plate	0.1	m	FRT properties
Specific heat capacity of polyurethane foam plate	871	J/kg·K	FRT properties
Thermal conductivity of polyurethane foam plate	0.020	W/m·k	FRT properties
Space size of simulation	4.1 × 2.2 × 2.2	m^3^	Clearance size
Vent speed	1–7	m/s	Control
Vent temperature	0	°C	Constant
Initial temperature in the body of the box	20	°C	Multipoint
External temperature	35	°C	Temperature
Air density	1.293	kg/m^3^	Air properties
Air-specific heat capacity	1006.43	J/kg·K	Air properties
Air thermal conductivity	0.0242	W/m·k	Air properties
Air kinetic viscosity	1.7894 × 10^−5^	kg/m·s	Air properties

**Table 2 foods-12-01837-t002:** Evaluation index of the temperature field to be measured.

Pattern	Average Deviation from the Temperature Difference Ratio (ψ_1_)	Inhomogeneity Coefficient Ratio (ψ_2_)	Maximum Difference Ratio (ψ_3_)	Temperature Field Uniformity Index (ξ)
Original state	1.74	7.49	0.89	10.12
Upper ventilation ducts	1.54	6.44	0.76	8.74
Lower ventilation ducts	1.69	5.38	0.54	7.61
Upper and lower ventilation ducts	1.38	4.39	0.36	6.13

## Data Availability

The datasets generated for this study are available on request to the corresponding author.

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
