# Peer review of "Evaluation of Temperature Uniformity in a Middle-Refrigerated Truck Loaded with Pig Carcasses"

_foods, 2023, doi:10.3390/foods12091837_

Round 1
Reviewer 1 Report
Dear Authors,
the figures presented very good informations; the pattern have been well chosen; the material and methods have been also well chosen; the model assumption and conclusion were good desribed;
I have some suggestion or questions:
The introduction does not fully describe the research topic undertaken. This introduction is the shortest introduction I've ever seen. It should be described in more detail based on the available literature;
In addition, the refrigerator truck was turned on for pre-cooling. To what temperature? Give the info in text;
What was the temperature of the pig carcasses loaded into the car? Give the info in text;
Pattern 5-12 - please give explanation for every sign;
3.3. what is FRP? This abbreviation is first time in section 3.3., and is not explained;
pattern 14: I understand that all signs are the same like in pattern 13? It may be good to make a table of abbreviations in one place of the publication, not in the publication's text, since abbreviations are repeated in various patterns. This is a suggestion;
"The heat conduction equation of the inner and outer walls of the compartment can be expressed as follows ....." why qc? - give explanaition like in this text "The average heat flux density qh of the refrigerated
truck compartment wall can thus be expressed as";
Pattern 27-28 and 39 (I think 29 - please correct this) - please give explanation for every sign, what is Dt, Dk, Gk, Gb, etc.;
"In Fluent 17" I don;t understand why fluent 17?
Figure 3: x=1.1 m - it is cental axis; but I think central axis should have 2 vales because the car container is 2.2 x 2.2 x 4.1 m; and 1.1 m is not the central axis;
Pattern 32 - please give explanation for every sign for example Vx....;
"As Table 2 shows, a smaller numerical value indicates a closer approximation to the ideal standard temperature field for the truck." Can You explain what do You mean?
Figure 5 - please give explanation for every sign for example "Exp(original)"; without this abbreviation I can't give an information of the text is correct.
Good luck!
Author Response
Response to Reviewer 1 Comments
Point 1: The introduction does not fully describe the research topic undertaken. This introduction is the shortest introduction I've ever seen. It should be described in more detail based on the available literature;
Response 1: Your point makes sense. There are hundreds of papers on simulating the environment of refrigerated trucks, and most of them are written very well. Considering that there is still a review article to be published later, not enough literature was cited in this article. Nevertheless, I have added three interesting papers and thanks for your reminders."
Point 2: In addition, the refrigerator truck was turned on for pre-cooling. To what temperature? Give the info in text;
Response 2: The pre-cooling temperature is 0℃, and the pre-cooling time is 24 hours. Considering the high turnover rate, sometimes this condition cannot be met. Therefore, this paper studies the potential quality and safety hazards of pig carcasses caused by the inhomogeneity of the temperature field of refrigerated vehicles without sufficient pre-cooling.
Point 3:What was the temperature of the pig carcasses loaded into the car? Give the info in text;
Response 3: This is a very interesting question. Under the condition of insufficient pre-cooling, the temperature difference between the upper part and the lower part of a single pig carcass is 2 ℃, and the average temperature of the pig carcass is 2℃. If time permits, I would like to simulate the temperature difference curve of a single pig carcass, which would be more interesting.
Point 4:Pattern 5-12 - please give explanation for every sign;
Response 4: Thank you for your reminder. I have added new explanation after Formula 12.
Point 5:3.3. what is FRP? This abbreviation is first time in section 3.3., and is not explained;
Response 5: Fiberglass reinforced plastic (FRP), sometimes called Coroplast, is fabricated by fiberglass grating manufacturers. FRP is mentioned in 2.1.
Point 6:pattern 14: I understand that all signs are the same like in pattern 13? It may be good to make a table of abbreviations in one place of the publication, not in the publication's text, since abbreviations are repeated in various patterns. This is a suggestion;
Response 6: This is a great suggestion. I will describe the Greek abbreviations in a table by unified way in future articles, which could save a lot of space.
Point 7:"The heat conduction equation of the inner and outer walls of the compartment can be expressed as follows ....." why qc? - give explanaition like in this text "The average heat flux density qh of the refrigerated truck compartment wall can thus be expressed as";
Response 7: The variable "qc" in the given sentence represents the heat flux density of the inner and outer walls of the compartment. "Heat flux density" is a measure of the amount of heat transferred per unit area per unit time. In this context, it represents the rate of heat transfer through the walls of the compartment due to conduction.
Point 8:Pattern 27-28 and 39 (I think 29 - please correct this) - please give explanation for every sign, what is Dt, Dk, Gk, Gb, etc.;
Dt, Dk, Gk, Gb
Response 8: Thank you for your correction. I have added new explanation after Formula 28.
Point 9:"In Fluent 17" I don;t understand why fluent 17?
Response 9: Fluent 17 means the version number 17 of Fluent purchased by our school. It is better to describe: In Fluent software. And I also added the source of C constants.
Point 10:Figure 3: x=1.1 m - it is cental axis; but I think central axis should have 2 vales because the car container is 2.2 x 2.2 x 4.1 m; and 1.1 m is not the central axis;
Response 10: Yes, you are right. It is possible that the central axis should have two values, as the car container is 2.2 x 2.2 x 4.1 m. I made a mistake. It should describe a z=1.1 longitudinal section. I also changed more clear middle-refrigerated truck drawings in Figure 1.
Point 11:Pattern 32 - please give explanation for every sign for example Vx....;
Response 11: Yes. We added explanation of Pattern 32.
η— Volume ratio of refrigerated vehicle load
Vx—Volume of pig carcasses loaded in a refrigerated truck
V0—headroom volume of refrigerated truck.
Point 12:"As Table 2 shows, a smaller numerical value indicates a closer approximation to the ideal standard temperature field for the truck." Can You explain what do You mean?
Response 12: Certainly! In Table 2, there are numerical values for different models used to simulate the temperature field for a truck. The "ideal standard temperature field" refers to the actual temperature distribution that would be expected for the truck in a perfect scenario, with no heat transfer or other factors affecting the temperature distribution. In other words, a smaller value means that the temperature distribution predicted by the model is more similar to the ideal temperature distribution that would be expected under perfect conditions. This suggests that the model is more accurate in predicting the actual temperature distribution for the truck.
Point 13:Figure 5 - please give explanation for every sign for example "Exp(original)"; without this abbreviation I can't give an information of the text is correct.
Response 13: Sure. Here are some common abbreviations used in this maniscript:
Exp(Original) : Data measured on the original unmodified refrigerated truck
Exp(Lower): Data measured on a refrigerated truck added lower ventilation ducts
Sim(Original): The simulated values on the original unmodified refrigerated truck
Sim(Lower): The simulated values on a refrigerated truck added lower ventilation ducts
Exp(Up): Data measured on a refrigerated truck added up ventilation ducts
Exp(Up&Lower): Data measured on a refrigerated truck added both up and lower ventilation ducts
Sim(Up): The simulated values on the refrigerated truck added up ventilation ducts
Sim(Up&Lower): The simulated values on the refrigerated truck added both up and lower ventilation ducts
In the context of the original text, "Exp(original)" likely refers to the original experimental data or measurements, as opposed to data obtained from simulations. As you menstioned, we added explanation under the Figure 5.

Reviewer 2 Report
Thank you for the invitation to review this interesting hot topic article. The article is well-written and gives a clear picture of the actual state of knowledge – the literature in this area was thoroughly reviewed.
The manuscript is comprehensive and meticulously written with a nice flow, making the topic easily readable. The chronology of the topic covered is also satisfactory. Additionally, the strength of this manuscript is the self-explanatory figures and a table.
Research paper (Title: Evaluation of Temperature Uniformity in a Middle Refrigerated Truck Loaded with Pig Carcasses) by Bai et al. have made an exhaustive effort to discuss different perspectives and approaches for data analysis. Text is clear and succinct in describing observations and conclusions. Design and implementation are adequate.
Author Response
Thank you for the invitation to review this interesting hot topic article. The article is well-written and gives a clear picture of the actual state of knowledge – the literature in this area was thoroughly reviewed.
The manuscript is comprehensive and meticulously written with a nice flow, making the topic easily readable. The chronology of the topic covered is also satisfactory. Additionally, the strength of this manuscript is the self-explanatory figures and a table.
Research paper (Title: Evaluation of Temperature Uniformity in a Middle Refrigerated Truck Loaded with Pig Carcasses) by Bai et al. have made an exhaustive effort to discuss different perspectives and approaches for data analysis. Text is clear and succinct in describing observations and conclusions. Design and implementation are adequate.
Many thanks to the reviewer for your encouragement. Nevertheless, I have revised some things that were not clearly described. At the same time, several references have been added to thank colleagues for their guidance.
Reviewer 3 Report
The manuscript is well prepared and presented. The trial plan is good and regularly studied. Calculations and methods used in calculations are chosen correctly. Obtained results are discussed correctly.
Author Response
The manuscript is well prepared and presented. The trial plan is good and regularly studied. Calculations and methods used in calculations are chosen correctly. Obtained results are discussed correctly.
Thanks again to the reviewer for your time. In order to thank the peers for their help, the authors added several references and corrected a little bit of wording to facilitate reading.